# Attentional focus strategies in racket sports: A systematic review

**Marcin Starzak**[1]\*, **Tomasz Niźnikowski**[1], **Michał Biegajło**[1], **Marta Nogal**[1], **Weronika Łuba Arnista**[2], **Andrzej Mastalerz**[3], **Anna Starzak**[1]

1 Faculty of Physical Education and Health in Biała Podlaska, Józef Piłsudski University of Physical Education in Warsaw, Biała Podlaska, Poland, 2 Faculty of Health Sciences, Lomza State University of Applied Sciences, Lomza, Poland, 3 Faculty of Physical Education, Józef Piłsudski University of Physical Education in Warsaw, Warsaw, Poland

\* marcin.starzak@awf.edu.pl

**Data Availability Statement:** All relevant data are within the manuscript and its Supporting information files.

**Funding:** The authors received no specific funding for this work.

## Abstract

The body of evidence has shown that the external focus of attention (EF) rather than the internal focus of attention (IF) enhances motor skill learning and performance. Within racket sports which require a high level of motor control, anticipation skills, and mental preparedness, effectively directing attention is essential to elicit improvements in athletic performance. The present review aimed to evaluate the scientific evidence concerning the effects of attentional focus instructions on motor learning and performance in racket sports. We systematically reviewed the literature according to the Preferred Reporting Items for Systematic Review and Meta-Analyses (PRISMA) guidelines. The study was registered with the Open Science Framework (osf.io/m4zat). Four electronic databases (Web of Science, Scopus, MEDLINE, and SPORTDiscus) were searched for original research publications. Inclusion criteria were: peer-reviewed journals; healthy and free from injury participants; attentional focus literature specific to the external or internal focus; attentional focus related to motor learning or motor performance; studies included at least one comparator (e.g., different attentional focus group, or control groups with neutral or no specific instruction); publications in which task(s) or skill(s) related to one of the racket sports (tennis, table tennis, badminton, squash, or padel). The initial search yielded 2005 studies. Finally, 9 studies were included in the quantitative analysis. Overall, the results indicated that EF benefits the learning and performance of racket sport skills, compared to IF and over control conditions. The findings suggest that coaches and practitioners should consider the adoption of EF to optimize racket skills performance, particularly in novice or low-skilled athletes.

## Introduction

Racket sports refer to physical activities using rackets to hit a ball or a shuttlecock. This group of sports includes tennis, table tennis, badminton, and squash. Some other racket sports are practiced to a lesser extent, e.g. padel or racquetball. The common characteristic of racket sports is that strokes are played by athletes in alternation. To succeed in these sports, athletes must possess a wide range of qualities and attributes. These include aerobic and anaerobic

**Competing interests:** The authors have declared that no competing interests exist.

fitness, speed, strength, agility, flexibility, motor coordination, mental toughness, technical and tactical skills, perception and action, awareness, and control [1,2]. Athletes must also possess exceptional motor control, which encompasses the ability to effectively control the ball or shuttlecock during strokes by appropriately manipulating a racket. Additionally, they must develop anticipatory skills [3], and be capable of swiftly adapting to the constantly changing game conditions, such as the wide range of flight, rotation, and speed of the ball or shuttlecock or the playing styles of their opponents. Another crucial characteristic of this group of sports is the required accuracy of all strokes played [3]. Therefore, the performance of racket sports is influenced by a complex interaction between technical, tactical, physiological, and psychological skills of players. In this regard, it is imperative for players to effectively focus their attention on tracking the ball or shuttlecock movement, anticipating its arrival, and ultimately hitting it toward the opponent's side.

Sports practitioners commonly use verbal instructions or feedback to effectively direct athletes' attention and improve performance and the learning process [4]. Effective cueing must direct the focus of attention to relevant information while performing motor skills. Athletes' attention can be directed in two main ways, either by focusing on the effect of their movement in the environment (external focus, EF) or by focusing on their body movement itself (internal focus, IF) [5]. Previous literature in the attentional focus field demonstrates the benefit of an EF relative to an IF across various tasks and populations. For instance, the research investigated movement effectiveness across sports skills with accuracy demands including dart throwing [6,7], golf shot [8,9], or Frisbee throwing [10]. The benefits of using EF over IF were also well reported for movement efficiency, as measured by jumping distance [11], sprint time [12], or muscle endurance [13].

Increased movement effectiveness and efficiency due to the EF are well presented in a theoretical framework of the constrained action hypothesis [14]. Although the exact mechanism of this theory is not fully understood, an increasing amount of evidence indicates that EF plays a crucial role in improving motor performance by facilitating a more automated control process. On the opposite, focusing internally allows for conscious motor control and constrained action, which results in reduced movement efficiency. Research confirms that the constrained action hypothesis provides a compelling explanation for why focusing externally leads to enhanced motor performance and learning.

Numerous research involving a control condition in their study designs without specific attentional instructions resulted in a similar effect as IF, and thus decreased performance relative to EF instructions [15]. Probably, in conditions with a lack of specific instruction, the attention intuitively shifts to the aspects related to the body movement [4]. Interestingly, some research suggests analogy instruction or holistic focus as an alternative to an EF [16]. Both represent such a kind of instruction that alters one's attention without any reference to the body, leading to less conscious control processing [17]. Therefore, available research results suggest that these instructions elicit similar benefits in motor performance over an IF [18]. Apart from the use of alternatives, EF itself may be distinguished between a proximal EF where attention is directed to aspects close to the body, or a distal EF with attention directed farther away from the body. Specifically, focusing on distal targets was shown to be more effective than proximal EF, resulting in a high degree of movement automaticity and efficacy [19,20].

Recently, the Ecological Dynamics Account of Attentional Focus introduced a novel explanation for the focus of attention effects, departing from the conventional constrained action hypothesis [21]. This framework is based on dynamical systems theory and ecological dynamics, where movement emerges from a dynamic process of self-organization in relation to organismic, environmental, and task constraints. According to the Ecological Dynamics Account of Attentional Focus, attentional focus is not a one-size-fits-all concept [22,23].

Instead, it depends on the specific constraints that influence perception for action. Both external and internal foci of attention offer unique benefits, but these advantages vary depending on the situational constraints. In essence, tasks that are guided by specific environmental information can be enhanced by adopting an external focus on relevant aspects within the environment. Conversely, tasks that are guided by specific bodily information can be improved by adopting an internal focus on relevant aspects concerning the body [21].

Racket sports are highly complex due to many variables that impact performance. One of the most demanding aspects of this sport is the development of mental skills, with attention control being particularly crucial [24]. Having the ability to effectively direct attention may significantly contribute to enhancing racket skill performance, particularly in movement accuracy and automaticity under environmental constraints. Optimizing training methods requires understanding how instructions for attentional focus affect motor learning and performance. However, there is a limited body of literature that shows a way of assisting coaches in organizing their thinking about effective attentional focus cues provided for learners in performing and developing racket sports skills. Therefore, this review aims to summarize the evidence on how attentional focus instructions affect motor learning and performance in racket sports.

## Materials and methods

The study systematically reviewed available evidence in accordance with the Preferred Reporting Items for Systematic Review and Meta-Analyses (PRISMA) guidelines. The study protocol was registered with the Open Science Framework (DOI: 10.17605/OSF.IO/34ZAV).

### Search strategy

A systematic search of the relevant literature was performed (December 2022) in the electronic databases Web of Science, Scopus, Medline (via EBSCO), and SPORTDiscus (via EBSCO) to identify articles published with no time restrictions. One of the authors (MS) conducted the initial search using keywords with Boolean operators "AND" and "OR" with various combinations in the title and/or abstract and/or full texts. The search requests were as follows: ("focus of attention" OR "attentional focus" OR "attentional foci" OR "external focus" OR "internal focus" OR "external foci" OR "internal foci" OR "attentional strateg*") AND ("tennis" OR "table tennis" OR "badminton" OR "squash" OR "padel"). We combined the search results and removed duplicates automatically using the management software EndNote X7 (Thomson Reuters, Philadelphia, PA, USA). All grey literature (e.g., conference abstracts, and unpublished data) were excluded from the final analysis. The flow chart of the systematic search process is summarized in Fig 1.

### Eligibility criteria

The studies inclusion criteria were as follows: 1) peer-reviewed journals; 2) healthy and free from injury participants; 3) attentional focus literature specific to the external/internal focus; (4) attentional focus related to motor learning and motor performance was used; 5) studies included at least one comparator (e.g., different attentional focus group, or control groups with neutral or no specific instruction); 6) publications in which task/s or skill/s related to one of the racket sports (tennis, table tennis, badminton, squash or padel) were studied. Studies were excluded if they were not published in English, reported as an abstract only, or not included sufficient data.

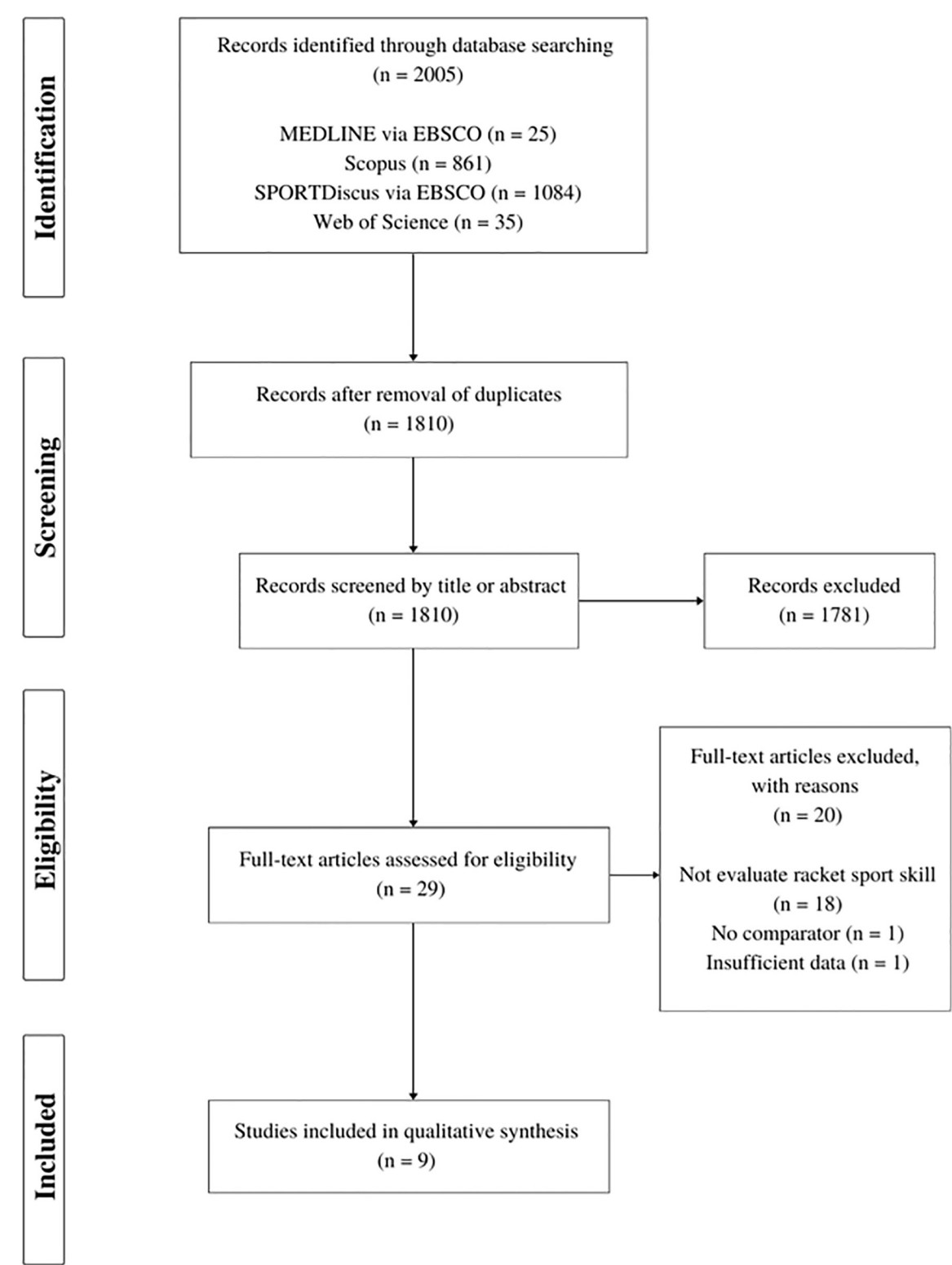

**Fig 1. The PRISMA flow diagram of the selection process.**

## Study selection

The study selection process was carried out by two reviewers (T.N., M.B.), who independently screened the titles and abstracts and analyzed the full texts. Any remaining duplicates were manually checked and removed. Any disagreement regarding study eligibility was discussed with the co-author (M.S.) for clarification.

## Data extraction

Two independent reviewers (M.N. and M.B.) extracted data, and any disagreements were resolved by discussion with a third author (M.S.). Data were extracted from each eligible study according to author and year of publication, racket sport, sample characteristics (level of expertise, gender, and mean age of participants), characteristics of intervention (protocol, type, and content of instruction), and main outcomes. The characteristics of the included studies are displayed in Table 2.

## Study quality and assessment of the risk of bias

Two review authors (M.S. and M.N.) assessed independently the possible risk of bias among eligible studies using the Physiotherapy Evidence Database (PEDro) scale [25]. The scale is a highly reliable and valid assessment tool for methodically evaluating the risk of bias [26,27]. The PEDro checklist contains 11 items, with 10 of them being scored, except for item 1, which refers to external validity. However, in studies that involve exercise training interventions, it is not feasible to blind the subjects and therapists (items 5 and 6), as they are actively participating in the exercise program, or it is almost impossible to blind therapists (item 7) [28]. Therefore, the total score ranges from 0 to 7 points. A point was awarded for each criterion the study met. The studies were categorized as "low" (PEDro scale 0–3), "moderate" (PEDro scale 4), "good" (PEDro scale 5), and "excellent" (PEDro scale ≥6). Points were awarded only when a given criterion was clearly met. We used the kappa correlation test to evaluate the agreement between the raters (M.S. and M.N.). If there were any ambiguous issues regarding rating points, a third author (T.N.) was included to reach a final consensus.

**Table 1. Characteristics of the included studies.**

| Study | Criterion | | | | | | | | | | | PEDro score |
|---|---|---|---|---|---|---|---|---|---|---|---|---|
| | 1 | 2 | 3 | 4 | 5 | 6 | 7 | 8 | 9 | 10 | 11 | |
| Abedanzadeh et al., 2022 [32] | 0 | 1 | 0 | 1 | 0 | 0 | 0 | 0 | 0 | 1 | 1 | 4 |
| Hadler et al., 2014 [33] | 0 | 1 | 0 | 0 | 0 | 0 | 0 | 0 | 0 | 1 | 1 | 3 |
| Keller et al., 2021 [34] | 0 | 1 | 0 | 1 | 0 | 0 | 0 | 0 | 0 | 1 | 1 | 4 |
| Koedijker et al., 2007 [30] | 1 | 1 | 0 | 0 | 0 | 0 | 0 | 0 | 1 | 1 | 1 | 4 |
| Mohamadi et.al., 2014 [35] | 1 | 1 | 0 | 1 | 0 | 0 | 0 | 0 | 0 | 1 | 1 | 3 |
| Niźnikowski et al., 2022 [36] | 1 | 1 | 0 | 1 | 0 | 0 | 0 | 0 | 0 | 1 | 1 | 4 |
| Tsetseli et al., 2016 [29] | 0 | 1 | 0 | 1 | 0 | 0 | 1 | 0 | 0 | 1 | 1 | 5 |
| Tsetseli et al., 2018 [37] | 0 | 1 | 0 | 1 | 0 | 0 | 0 | 1 | 0 | 1 | 1 | 5 |
| Wulf et al., 2000 [31] | 0 | 1 | 0 | 0 | 0 | 0 | 0 | 0 | 0 | 1 | 0 | 2 |

PEDro rating criteria (1) eligibility criteria were specified, (2) subjects were randomly allocated to groups, (3) allocation was concealed, (4) the groups were similar at baseline regarding the most important prognostic indicators, (5) there was blinding of all subjects, (6) there was blinding of all therapists who administered the therapy, (7) there was blinding of all assessors who measured at least one key outcome, (8) measures of at least one key outcome were obtained from more than 85% of the subjects initially allocated to groups, (9) all subjects for whom outcome measures were available received the treatment or control condition as allocated, (10) the results of between-group statistical comparisons are reported for at least one key outcome, (11) the study provides both point measures and measures of variability for at least one key outcome.

**Table 2. Characteristics of the included studies.**

| Study | Sport | Participants | | Task | Intervention | | Outcomes |
|---|---|---|---|---|---|---|---|
| | | Level | Sex (n), mean age (in years) | | Protocol / Duration | Instructions / Manipulation check | |
| Abedanzadeh et al., 2022 [32] | Badminton | Novice, physical education students | M = 60, 19.57 ± 0.98 yrs, range 15–23 yrs | Badminton short serve | 5 days, 3 x 10 trials (150 total trials) | EF: "focus on the movement of the racquet during the serve." HF: "focus on feeling smooth and fluid when completing the serve." IF: "focus on the movement of your arm during the serve." CON: no focus cue Manipulation check: yes | Accuracy test: *AQ*: EF↑, HF↑, IF↑, CON↑; HF>IF, CON; EF = HF = IF, IF = CON *RT*: EF, HF>CON; EF = IF = HF, IF = CON *TT*: HF>IF, CON; HF = EF, EF = IF = CON |
| Hadler et al., 2014 [33] | Tennis | Novice, children | F = 21, M = 24, 10.98 ± 0.72 yrs | Forehand drive | 60 practice trials | EF: "focus on the movement of the racquet", IF: "focus your attention in the movement of your arm", CON: no focus instructions Manipulation check: no | Accuracy test: *AQ*: EF↑, IF↑, CON↑ *RT*: EF>IF; IF = CON *TT*: EF> IF; IF = CON |
| Keller et al., 2021 [34] | Tennis | High level, young tennis players | M = 10, 19.2 ± 2.7 yrs | Tennis serve | 100 serves (5 conditions x 2 sets x 10) | IF: "serve as fast as possible while landing the serve in the target zone by accelerating your arm as fast as possibles" EF: "serve as fast as possible while landing the serve in the target zone by accelerating your racket as fast as possibles" AF: "serve as fast as possible while landing the serve in the target zone and try to maximize the speed shown on the screen" AF+EF: "serve as fast as possible while landing the serve in the target zone by accelerating your racket as fast as possible and by trying to maximize the speed shown on the screen" CON: "serve as fast as possible while landing the serve in the target zones" Manipulation check: no | Service speed: AF>EF, CON Serves in the target zone: EF = IF = AF = AF +EF = CON |
| Koedijker et al., 2007 [30] | Table tennis | Novice, adults with little or no experience | M = 11, F = 31, 21.8 ± 3.6 yrs | Forehand drive | 9 blocks x 50 trials | EL: explicit set of instructions about how to execute the forehand IL: only a single instruction in the form of an analogy EF: instruction to attend the ball at all time IF: instructions to specifically focus on movement execution Manipulation check: no | Accuracy and quality assessment: *AQ*: EL↑, IL↑, EF↑, IF↑ EL>IL, IF, EF *RT*: IL>EL = EF>IF *TT*: IL>EL, IF, EF |

(*Continued*)

**Table 2.** (*Continued*)

| Study | Sport | Participants | | Task | Intervention | | Outcomes |
|---|---|---|---|---|---|---|---|
| | | Level | Sex (n), mean age (in years) | | Protocol / Duration | Instructions / Manipulation check | |
| Mohamadi et. al., 2014 [35] | Table tennis | Novice, sport high school students | F = 80, 16.6 ± 0.6 yrs | Backhand drive | 6 sessions (2/wk, 3 sets x 10 trials) | IF: "slightly rotation" ST<br>EF-n (near):"slightly open" ST<br>EF-d (distal)"over the net" ST<br>EF-i (increase in distance):"slightly rotation, slightly open, and over the net" ST<br>CON: no ST<br>Manipulation check: yes | Accuracy assessment:<br>*AQ*: EF-i↑, EF-d↑, EF-n↑, IF↑;<br>EF-i, EF-d, EF-n, IF>CON<br>*RT*: EF-i, EF-d, EF-n>IF<br>*TT*: EF-i, EF-d, EF-n> IF<br>Movement assessment:<br>*AQ*: EF-i↑, EF-d↑, EF-n↑, IF↑;<br>EF-i, EF-d, EF-n, IF>CON<br>*RT*: EF-i, EF-d, EF-n>IF<br>*TT*: EF-i, EF-d, EF-n> IF |
| Niźnikowski et. al 2022 [36] | Table tennis | Low-skilled, undergraduate physical education students | F = 12, M = 39, 22.9 ± 1.8 yrs | Backhand drive | 3 blocks x 15 trials | IF: "concentrate on the hand holding the paddle"<br>EF-p (proximal): "concentrate on the ball"<br>EF-d (distal): "concentrate on targets marked on the tennis table"<br>Manipulation check: no | Accuracy assessment:<br>*AQ*: EF-p↑, EF-d↑, IF =;<br>EF-d>IF<br>*RT*: EF-p↑, EF-d↑, IF↑;<br>EF-d>IF |
| Tsetseli et al., 2016 [29] | Tennis | Novice, children All players had 1–2 yrs (M = 1.5 ± 0.4) experience | F/M = 60, 8.4 ± 0.5 yrs | Tennis serve, forehand drive, backhand drive | 6 wks, 2/wk | IF: five different IF instructions<br>EF: five different EF instructions<br>CON: no attentional focus instructions<br>Manipulation check: no | Decision making:<br>*AQ*: EF↑; IF =; CON =;<br>EF>IF; EF>CON; IF = CON<br>*RT*: EF>IF; EF>CON; IF = CON<br>Skills execution:<br>*AQ*: IF↑; EF =; CON =;<br>EF>CON; IF = EF, IF = CON<br>*RT*: EF>CON; IF = EF, IF = CON<br>Base position:<br>*AQ*: IF↑; EF↑; CON↑;<br>EF>CON; IF = EF, IF = CON<br>*RT*: EF>CON; IF = EF, IF = CON<br>Total game performance:<br>*AQ*: IF↑; EF↑; CON↑;<br>EF>CON; IF = EF, IF = CON<br>*RT*: EF>CON; IF = EF, IF = CON |

(*Continued*)

**Table 2.** (Continued)

| Study | Sport | Participants | | Task | Intervention | | Outcomes |
|---|---|---|---|---|---|---|---|
| | | Level | Sex (n), mean age (in years) | | Protocol / Duration | Instructions / Manipulation check | |
| Tsetseli et al., 2018 [37] | Tennis | All players had 1 year experience (M = 1.2 ± 0.6) | M/F = 68, range 8–9 yrs (M = 8.8 ± 0.54), | Tennis serve, forehand drive, backhand drive | 6 wks, 2/wk | IF: five different IF instructions EF: five different EF instructions CON: no attentional focus instructions Manipulation check: no | Technical assessment of service, forehand, and backhand: *AQ*: EF>IF, CON; IF = CON *RT*: EF>IF, CON Performance score: *AQ*: EF>IF, CON; IF = CON *RT*: EF>IF, CON |
| Wulf et al., 2000 [31] Experiment 1 | Tennis | Novice, adults | M = 15, F = 21, range 16–33 yrs | Forehand drive | 2 practice sessions: 10 x 10 trials, 3 x 100 trials | EF-a (antecedent): "focusing on the ball coming toward them EF-e (effect): "focusing on the ball leaving the racket Manipulation check: no | Performance score: *AQ*: EF-a↑, EF-e↑; EF-a = EF-e *RT*: EF-a↑; EF-e↑; EF-e>F-a |

AF–augmented feedback, EF–external focus group, IF–internal focus group, HF–holistic focus group, CON–control condition, *AQ*–acquisition, *RT*–retention test, *TT*–transfer test, ST–self-talk.

## Results

### Study selection

The initial search made in the databases retrieved 2005 titles, of which 1781 were excluded based on titles, abstracts, duplicate studies and other reasons (Fig 1). Twenty-nine potentially relevant studies were identified for full-text analysis. Finally, 9 studies were selected for the qualitative synthesis. Agreement between the reviewers (M.N. and M.B.) on abstract and full-text screening was, with a kappa score of 0.93 and 0.97, respectively.

### Risk of bias and quality assessment of studies

The results of the risk of bias in the included studies are presented in Table 1. The quality score of the PEDro scale ranged from 2 to 5 (mean 3.8 ± 1.0). The kappa agreement between the reviewers was 0.95 showing almost perfect agreement. Most of the studies assessed showed a high risk of bias. Accordingly, two studies were classified as good quality (score 5), four studies were defined as studies with moderate quality (score 4), and three were defined as studies with poor methodological quality (score 3 or below). Except for one study [29], none of the included studies reported the criteria of blinding methods (subjects, therapists, and assessors), concealed allocation, and completeness of follow-up. Only in one study [30] was it specified that the subjects were analyzed according to their initial group allocation. Moreover, except for one study [31], all the studies also provided both point measures and measures of variability for at least one key outcome.

### Study characteristics

Table 2 summarizes the characteristics of the extracted studies. Five of nine studies focused on the effects of attentional focus in tennis [29,31,33,34,37], three were conducted in table tennis [30,35,36], and one study in badminton [32]. No study was found for squash and padel.

In terms of the motor tasks under investigation, generally three basic skills in racket sports were used, including service, forehand, and/or backhand drive. Two of the studies [32,35] included a manipulation check in the experimental design to measure participants' accuracy and consistency in following the prescribed attentional focus instructions. The majority of studies employed specific accuracy tests to evaluate performance and measure learning effectiveness. Furthermore, in other studies, the main outcomes were skill execution score, service speed, movement assessment, or total game performance based on components of decision-making, skills execution, and Base.

## Participant characteristics

The sample sizes for particular studies ranged from 10 to 80 participants, with a total of 452. Most of the studies combined male and female participants within methodological design, with two studies restricting their sample to males, and one study to women only. For one study, gender distribution was not reported. Age ranges from 8 to 33 years, so that includes children to adults. Regarding the level of expertise in most studies, the subjects were novices or subjects with little experience in racket-type sports training. Out of the nine studies, three included a sample size that comprised highly skilled participants. In a particular study [34], the participants were national-level tennis players. However, the authors did not specify their experience. Two additional studies [29,37] included young tennis players who had a minimum of one year of experience.

## Intervention characteristics

All studies that assessed the impact of learning used both a pre- and a post-test design, measuring the variables before and after the intervention sessions. The duration of the intervention period ranged between one session and 6 weeks. Only one study [34] examined the immediate effects on tennis service performance when adopting attentional foci, augmented feedback, or a combination of augmented feedback with EF. Five studies performed a follow-up retention test [29,33,35,36,38], and in two studies [33,35] a transfer test was also used.

One study used only a single intervention training period in tennis [33], and two in table tennis [30,36], respectively. One study [38] examining the effects of attentional instructions in tennis consisted of two intervention sessions.

In two experiments [35,36], the impact of EF was tested with respect to the direction of the body (proximal, distal, or increasing in distance). However, in Mohamadi et al. [35] the focus of attention was applied through instructional self-talk.

Most of the studies used experimental and control comparisons, where the control group did not receive any attentional focus instruction, or the participants received neutral cues. Other study designs contained two or more experimental groups to compare the effects of the training intervention.

## Descriptive synthesis

Table 2 presents a summary of the intervention variables and key findings from the 9 studies that were reviewed. We compared the main outcomes of the attentional focus interventions to provide a comprehensive overview.

**Effects of EF versus IF and control conditions.** Most of the studies revealed a significant improvement in the main outcomes for all treatment groups from the initial to the final measurement across the acquisition phase. An EF was more beneficial than an IF in the post-test for decision-making [29], technique development [37], and accuracy performance in tennis [33,35,36]. Several studies showed improvement with an EF compared to control conditions in

the post-test regarding the accuracy task assessment [32,33], decision-making skill [29], and total game performance score [37].

The retention test was conducted in the majority of studies, except for the study of Mohamadi et al. [35]. In two of the included studies in this review [29,37] the authors performed similar retention protocols to assess the effects of attentional focus on learning. The examination of the retention test was made after a delay of one week under the same conditions as during the acquisition phase. Tsetseli et al. [29] studied whether EF or IF affects game performance, including decision-making, skills execution, and Base. The participants in the EF group showed significantly higher performance in decision-making compared to the participants of the IF group and the control group, respectively. They also found that the EF group resulted in better total game performance, skill execution, and Base evaluation than the control group but not over the IF group. No significant differences were revealed between the IF and the control group, in any aspect of game performance.

Tsetseli et al. [37] reported that participants who received external instructions instead of focusing internally or being in control conditions showed significant improvements in their technical assessment of the forehand, backhand, and service. There were also no differences between IF and control conditions across all outcome measures.

Hadler et al. [33] carried out a retention test two days after the intervention period using the same task that was practiced and with no attentional instructions or reminders to participants. The accuracy of forehand tennis stroke with the dominant arm was improved when participants were instructed to adopt an EF compared to both IF and control conditions. The IF and control group performance did not differ.

Furthermore, two studies of the impact of attentional focus on learning effects were also measured by the transfer test. Hadler et al. [33] conducted a transfer test involving the execution of a novel task (different distance and target zone) two days after the intervention period with no focus instructions. They found that the EF group demonstrated higher accuracy scores of forehand tennis drive compared to the IF group. There were no differences between the EF group and control conditions, as well as between IF and control conditions.

In another study [32] a transfer test was administered 30 minutes after the completion of the retention phase and involved tennis service from the opposite side of the court compared to conditions during the practice and retention test. The participants did not receive any attentional focus cues. The results indicated that the EF group did not significantly differ from the IF group.

Finally, two studies [31,36] investigated the effects of various attentional focus instructions without a control group that received no instructional cues. The study by Niźnikowski et al. [36] found that both external foci groups (distal and proximal) improved performance from the pre- to post-test, but there was no improvement in the IF group. Moreover, both EF groups demonstrated equal effectiveness when compared to the IF group in the retention test. In the second research [38] the external instruction focused on the oncoming ball and external instruction focused on the ball leaving the performer's racket were investigated. After the intervention period, both groups demonstrated significant improvements in accuracy performance. It was also observed that during the retention test, participants achieved significantly better performance outcomes when they focused their attention on the effects of their movement rather than the antecedent of the performer's action.

**Effects of alternative focus strategies on performance and learning.** Three of the nine studies included in the final analysis indicated that alternative forms of instruction may yield superior movement outcomes compared to EF. Abedanzadeh et al. [32] demonstrated that a holistic focus of attention provided a performance benefit relative to the control group during acquisition and a learning benefit over the internal and control group during the transfer test

in the badminton accuracy service test. The findings also showed that an EF group acquired a learning benefit compared to the control condition in retention and a marginal benefit over control during acquisition. Moreover, there were also no differences between the EF and IF group.

In another study, Koedijker et al. [30] showed comparable learning outcomes in both the EF and IF conditions. The retention and transfer tests were conducted immediately following the last practice trial. The group that received a single instruction through an analogy showed improved performance compared to the other conditions during the high-pressure test and dual-task test.

Keller et al. [34] conducted a study to investigate the impact of attentional instructions, augmented feedback, and their combination with EF on the immediate performance of service speed and accuracy. They found that the adoption of an EF did not result in faster serves compared to an IF or the control group. Similarly, the combination of augmented feedback and EF did not result in any additional improvement in service speed across conditions.

## Discussion

This systematic review aimed to determine the effects of adopting attentional instructions on the learning and performance of various racket sports skills. The 9 included studies involved different samples from different racket sports, varying in study design, intervention variables, and main outcomes. Overall, the results of this review indicate that using an EF has advantages for learning and performance of racket sport skills compared to an IF or control conditions.

### General findings

The main findings of this systematic review are consistent with a substantial body of evidence that confirms the beneficial effects of an EF on both motor learning and performance [39]. Chua et al. [39] conducted an exceptionally comprehensive review on the attentional focus, synthesizing the findings from over two decades of research in this area. The results of their study revealed the conclusive advantages of employing EF instead of IF for enhancing motor learning and performance, irrespective of age, health, or level of expertise [39]. Similarly, numerous other review studies support the notion that implementing an EF results in superior motor learning and performance compared to an IF [15,40]. However, according to a recent umbrella review conducted by Werner et al. [41], there is insufficient evidence to support the effectiveness of implementing EF instruction for technique training in sports. To make precise adjustments to movement techniques, it is recommended to consider the utilization of attentional cues, specifically through the utilization of IF instructions. Furthermore, a recent Bayesian meta-analysis challenges the prevailing consensus that an external focus is superior to an internal focus. The study indicates that previous meta-studies of the attentional focus may have been influenced by reporting bias. The analysis found significant unexplained heterogeneity in the effects of attentional focus, which indicates that the impact of attentional focus may vary depending on situational factors or methodological issues. It is uncertain whether an IF may be equally or even more effective than an EF in many scenarios. Additionally, it is difficult to identify the specific sources of heterogeneity in the effects of attentional focus. As a result, the effects of EF tend to be minimal or even non-existent [42].

This review supports the existing knowledge in the field of attentional focus, emphasizing the benefits of an EF over an IF. In most studies, adopting instructions that focus on the movement effect enhanced outcome effectiveness such as accuracy in forehand drive, compared to conditions when the attention was focused on the movement itself. It has been suggested that focusing internally on one's movements constrains the automatic control of the motor system

and may lead to disrupted accuracy in throwing and aiming performance [42–44]. However, most studies did not precisely assess the automatic control of the motor system, such as EMG activity or movement fluency, which are indicators that an EF of attention leads to more automatized movements than an IF [45]. Therefore, we cannot fully support the argument that an external focus promotes automatic control of actions, thus preventing the motor system from being constrained by conscious cognitive control for racket skills performance compared to an internal focus. Future studies should consider these factors to provide a comprehensive understanding of the relationship between attentional focus and motor performance in racket sports skills.

It should be noted that many research on attentional focus also reported that an EF is superior to both an IF and a control condition without focus [15]. Majority of the studies in this review included control conditions without any attentional instructions. Mostly, control conditions resulted in similar performance or learning outcomes as IF conditions, with both being less effective than EF conditions. Therefore, this could be also evidence of the beneficial effects of adopting instructions related to the movement effect rather than detrimental effect of a conscious control through internally focusing on the body movement [15].

Among the included studies, the potential benefits of an EF on motor learning and performance were reported across skill levels, age, and sex. Collectively, the results demonstrate the advantage of an EF compared with an IF irrespective of participants' characteristics. Such benefits were previously well verified across a diversity of skill experience and populations [39]. However, these findings are in contrast with the findings of Keller et al. [34] who reported no differences between an EF and IF on service speed in elite tennis players. The authors pointed out that one of the possible explanations relates to a high level of participants' expertise for whom the performance-enhancing effect of external effect may be limited [46]. Secondly, the authors argue, that despite both instructions referring to task-relevant information with accordance to those described in other studies, only one word differed between them, which surprisingly did not produce the predicted effects. It is clear that the generalizability of the focus of attention effect in highly skilled athletes in racket sports requires further investigation.

## Attentional focus strategies in racket sports

Considering the findings of the current study, an external instruction's specificity was shown to impact the movement performance and learning process. Generally, in most of the reviewed investigations, in order to elicit an EF, the participant's attention was directed to the movement outcome (e.g. "focusing on the ball leaving the racket"), or components of the racket being held or the racket itself (e.g. "place your hand on the red mark on the grip") [29]. These instructions directing attention on a near aspects of the object or implement and have been viewed as an advantage when executing a variety of skills [33,47,48].

There were also studies where the instructions were directed at targets farther from the participants' body e.g. "concentrate on targets marked on the tennis table". For example, Mohamadi et al. [35] and Niźnikowski et al. [36] studied the impact of EF direction compared to IF or control instructions. In both investigations, the conditions with proximal and distal EF resulted in greater outcomes over the IF. Furthermore, regardless of the focus direction (proximal, distal), the impact on movement performance was equally effective for acquisition and retention. However, these findings are partly in line with well-established evidence of distance effect in providing an EF [39]. Generally, focusing on distal aspects of the performer's body compared to directing attention closer to the body leads to a greater performance improvement [39]. Concentrating on a more distal target makes the movement effect more easily distinguishable from the body movements that create the effect than concentrating on a more

proximal target [19]. According to recent work by Singh and Wulf [47] the effectiveness of focus direction might depend on the level of expertise. They concluded that the low-skilled performers may benefit more from a proximal focus than the high-skilled performers who may benefit from the distal focus. The theoretical framework of ecological dynamics further supports these findings. When novices focus on proximal aspects, such as racket motion, they attune to specific perceptual information and assemble optimal coordination patterns. Experts can exploit this information to enhance their motor automaticity by focusing distally (e.g. intended ball trajectory) in a late stage of learning [21]. Accordingly, additional research is necessary to better understand how focus direction affects learning and performance of racket sports, depending on skill expertise, and to draw conclusions.

Also other types of attentional focus strategies were employed to explore the potential impact on performance and learning relative to EF or IF instructions. Abedanzadeh et al. [32] tested whether both a holistic focus and an EF would be beneficial in learning and performance of a badminton service in novice compared to directing attention internally or without specific cue. The holistic focus of attention served a general feeling associated with completing a movement and involved a "focus on feeling smooth and fluid when completing the serve". They found that a holistic focus of attention provided a performance improvement relative to the control group during acquisition, and a learning benefit over the IF and control group during the transfer phase. Surprisingly, regarding the assumptions of previous studies, no differences were observed between directing attention externally and internally. On most analyzing blocks, the EF group appeared to have higher accuracy scores in the short serve task than the IF group, but these differences were not significant. The authors attribute these results to a possible bias that may arise from the sensitivity of the measure used to assess accuracy as a dependent variable.

Finally, similar to the holistic focus instructions, a form of analogy or metaphor might produce effects similar to those of EF. A holistic focus emphasizes the overall feeling generated by a movement, rather than the specific movements that create the performance outcome. This approach may lead to a higher level of automaticity than internal focus. Because holistic attention instructions are less specific, they appear to be more beneficial than focusing on controlling some movements in both skilled athletes and novices [49,50]. Similarly, the analogy has been recognized as a method for eliciting implicit processes during skill acquisition [51]. As analogies minimize the potential for disrupting movement action through conscious processing, they enable learners to draw inferences about concepts with minimal conscious effort. Such instructions result in switching the performer's attention from their movements to the movement goal, promoting a more automatic control process [18,51–53]. This is crucial for sports where there is no equipment or objects to focus on, and the participants tend to shift their attention to the body. However, providing instructions such as "move your racket as you want to make a small circle" [29] or "pretend to draw a right-angled triangle with the bat . . . and move the bat backwards over the bottom of the triangle and hit the ball while moving the bat upward along the hypotenuse" [30] can also be viewed as an advantage over IF conditions during racket skill execution. Therefore, improving learning and performance of racket sports skills could benefit from the use of holistic focus and analogies. Nonetheless, further studies are needed to strengthen this evidence.

The methodological quality of the studies included in this review varied from low to good, which is comparable to other reviews in the field of attentional focus where the majority of published studies are generally of medium quality [54]. However, it is worth noting that attentional manipulation yielded similar effects regardless of the methodological quality of the included studies. Whether the studies were considered low [33] or good [29,37] quality, the use of EF instructions had similar effects on performance outcomes compared to IF or control

conditions. Establishing reliable methodological procedures that ensure high internal validity is a challenge, and it affects the quality of published studies [28]. To improve the rigor of future studies, researchers must consider certain factors such as blinding assessors, effective allocation concealment, and complete follow-up.

Although all the studies provided a clear description of the intervention, only two of them [32,35] included an analysis that acknowledged the implementation of a manipulation check in their experimental design. However, the study conducted by Mohamadi et.al. [35] did not provide any descriptive results regarding adherence to attentional focus. In attentional focus research, a manipulation check is utilized to determine whether participants adhere to the given instructions for attentional focus under the specified testing conditions [55–57]. The absence of a manipulation check limited the possibility of measuring the accuracy and consistency with which participants followed the prescribed attentional focus instructions. Adopting a manipulation check procedure could be especially helpful in order to better understand how participants focus their attention when they are performing in control conditions following a neutral set of instructions [58]. Concerns regarding the impact of different foci of attention on the effectiveness of manipulating participants' focus may limit the certainty of conclusions in this review.

## Study limitations and future research

This review has some limitations that must be acknowledged. Despite a wide search of the four relevant databases, the grey literature was not examined. The search was also limited to journal articles published in English. Therefore, we are aware that some relevant references may have been missed out.

The studies in the present review were highly heterogeneous concerning the nature of the interventions and moderator variables, such as gender or skill levels which may influence the reliability of the results. This diversity of the studies was also a reason why the quantitative analysis of the results could not be conducted. Furthermore, the findings from this review must be interpreted with caution, with respect to the high risk of bias of studies.

The results of our systematic review encourage further studies on the attentional focus in racket sports (for all skill levels) both in young and adolescents. Further research should comprehensively evaluate attentional focus effects within a real sport context. Various sport-specific factors that could influence attentional focus, such as game dynamics or environmental conditions, need to be considered [29,37]. Conducting research in a real sports environment allows researchers to capture subtle aspects of attentional focus that may not be apparent in a controlled laboratory setting.

In addition, studying the impact of attentional focus at different stages of learning could provide valuable insight into the optimal strategies for skill acquisition and performance enhancement. Since most studies involved in this review were limited to a short duration of practice intervention that corresponds to the early stage of motor learning, it is crucial to conduct research that examines the extended impact of attentional focus on motor skills over an extended period of time [59]. There is a need for further evidence to better understand how or whether EF affects performance and learning at different stages of learning.

## Conclusions

The results indicate a positive effect on skill acquisition following the implementation of EF within racket sports can be supported. The majority of studies included in qualitative synthesis showed that directing attention to the external sources around the body (near or distal) rather than how it is produced (IF) can enhance performance and learning basic skills in racket

sports. The findings suggest that coaches and practitioners should consider the adoption of an EF of attention during the training of racket sport skills particularly in novice or low-skilled athletes.

## Supporting information

**S1 Checklist. PRISMA 2020 checklist.**
(DOCX)

## Author Contributions

**Conceptualization:** Marcin Starzak, Tomasz Niźnikowski, Michał Biegajło, Weronika Łuba Arnista, Andrzej Mastalerz, Anna Starzak.

**Data curation:** Marcin Starzak, Tomasz Niźnikowski, Michał Biegajło, Marta Nogal, Anna Starzak.

**Formal analysis:** Marcin Starzak, Tomasz Niźnikowski, Michał Biegajło, Marta Nogal, Anna Starzak.

**Funding acquisition:** Marcin Starzak, Tomasz Niźnikowski.

**Investigation:** Marcin Starzak.

**Methodology:** Marcin Starzak, Tomasz Niźnikowski, Michał Biegajło, Marta Nogal, Weronika Łuba Arnista, Andrzej Mastalerz, Anna Starzak.

**Project administration:** Marcin Starzak, Tomasz Niźnikowski, Marta Nogal.

**Resources:** Marcin Starzak, Tomasz Niźnikowski, Marta Nogal.

**Software:** Marcin Starzak.

**Supervision:** Marcin Starzak, Tomasz Niźnikowski, Michał Biegajło, Marta Nogal.

**Validation:** Marcin Starzak, Tomasz Niźnikowski, Michał Biegajło, Marta Nogal, Weronika Łuba Arnista, Andrzej Mastalerz, Anna Starzak.

**Visualization:** Marcin Starzak, Tomasz Niźnikowski, Michał Biegajło, Marta Nogal, Weronika Łuba Arnista, Andrzej Mastalerz, Anna Starzak.

**Writing – original draft:** Marcin Starzak, Tomasz Niźnikowski, Michał Biegajło, Marta Nogal, Weronika Łuba Arnista, Andrzej Mastalerz, Anna Starzak.

**Writing – review & editing:** Marcin Starzak, Tomasz Niźnikowski, Michał Biegajło, Marta Nogal, Weronika Łuba Arnista, Andrzej Mastalerz, Anna Starzak.

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
