## [Decision Letter · Decision Letter 0]

18 Jul 2023

PONE-D-23-11299Attentional focus strategies in racket sports: A systematic reviewPLOS ONE

Dear Dr. Niźnikowski,

Thank you for submitting your manuscript to PLOS ONE. After careful consideration, we feel that it has merit but does not fully meet PLOS ONE’s publication criteria as it currently stands. Therefore, we invite you to submit a revised version of the manuscript that addresses the points raised during the review process.

We look forward to receiving your revised manuscript.

Kind regards,

Nick Fogt

Academic Editor

PLOS ONE

Additional Editor Comments 

Both reviewers have substantial numbers of comments, all of which will need to be addressed.

Of particular note, reviewer #2 raises the issue of what sets this review apart from another recent review. Please fully address this question And most importantly, reviewer #2 makes an important point about just how much overlap there is between internal focused and external focused instructions, and whether investigators in previous studies can actually tell whether subjects focused in the manner asked of them by the investigators. These are very important issues to be addressed.

Reviewers' comments:

Reviewer's Responses to Questions

**Comments to the Author**

1. Is the manuscript technically sound, and do the data support the conclusions?

Reviewer #1: Yes

Reviewer #2: Partly

2. Has the statistical analysis been performed appropriately and rigorously? 

Reviewer #1: N/A

Reviewer #2: N/A

3. Have the authors made all data underlying the findings in their manuscript fully available?

Reviewer #1: Yes

Reviewer #2: Yes

4. Is the manuscript presented in an intelligible fashion and written in standard English?

Reviewer #1: No

Reviewer #2: Yes

5. Review Comments to the Author

Reviewer #1: Major Comments – Abstract

1. Pg. 2 Line 45: Does “paddle” refer to “padel” or “paddle tennis”? If it refers to the former, use “padel” to avoid ambiguity because it is the more popularly used term, and to be consistent with its usage elsewhere.

Major Comments – Method

Pg. 5 Line 117: Figure 1 mentioned “other sources”, but the search for relevant articles other than the use of the four electronic databases has not been described under this subsection on search strategy.

Pg. 5 Line 124: See comment on abstract regarding “paddle”.

Pg. 5 Line 128: How was the de-duplication process “automatic”?

Pg. 6 Line 133: Clarify “level”.

Pg. 6 Line 140: Since no scoring system has been described, it is unclear what “total score” means.

Pg. 6 Line 142: Who does “they” refer to?

Pg. 6 Lines 145‒146: The kappa coefficient is a statistic and is obtained using a type of correlation statistics known as the interrater reliability test. This test assesses the level of agreement amongst raters; it is not able to influence (control for) risk of bias.

Major Comments – Results

Pg. 7 Line 153 and 159: It is unclear if the same type of agreement analysis was conducted to evaluate the interrater reliability for abstract screening, for full-text screening, and for risk of bias assessment, or whether the same number of raters (reviewers) were used in the assessment, because either absolute or mean (rather than the same type of) kappa coefficient values were reported.

Pg. 8 Line 180: See comment on abstract regarding “paddle”.

Pg. 8 Line 188: Does “452” refer to the total number of participants combined across the nine included studies? There is no need to specify “men and women” because it would be clear from the subsequent sentences that the participants consisted of both males and females in the children and adult age groups.

Pg. 8 Line 191: Why would there be seniors as young as 33 years of age? Remove the comma.

Pg. 8 Lines 195‒196: Clarify the context for “practice experience”. How is it different from “training experience” or simply “experience”?

Pg. 9 Line 200: For readers who are not so familiar with the motor learning literature, an elaboration on the “immediate performance effect” may be required.

Pg. 9 Lines 207‒208: There is conceptual incongruence between “direction” and “proximal, distal or increasing in distance”. That is, it is unclear why this list was provided in parentheses as examples for “direction”.

Pg. 10 Lines 222‒223: Were these improvements found at a particular assessment timepoint (e.g., post-test, transfer test) or multiple timepoints?

Pg. 10 Line 228: What is “base”?

Pg. 10 Lines 229‒230: Provide the contextual clarification for “directed externally”. The comparative use of “participants” versus “athletes” in the same sentence seems to imply that there could be a mixture of non-athletes and athletes in the sample, and that they were not randomly assigned to the various treatment groups.

Pg. 10 Line 231: Does “base evaluation” refer to the baseline assessment administered before the start of the acquisition (practice) phase?

Pg. 13 Lines 256‒258: Rephrase so that it is clear that motor performance improvements did not differ between the distal EF and proximal EF groups, but both groups performed better than the IF group, for both the post-test and retention test.

Pg. 13 Lines 258‒260: What is the “second research”? The two “external instructions” should be described in a way that shows the differentiation between them.

Pg. 14 Line 269: Were these “findings” from Abedanzadeh et al. or the three studies referenced in the first sentence of this paragraph?

Pg. 273 Line 273: Retention and transfer tests are typically administered with a delay of at least 24 hours after the last practice trial. Does “immediately” mean these tests occurred within the same day as the last practice session?

Pg. 14 Line 279: It is unclear which condition(s) the combined effects of augmented feedback and EF were not better than.

Major Comments – Discussion

Pg. 14 Lines 281‒282: According to the review aim described on Pg. 3, both learning and performance effects were of interest.

Pg. 16 Lines 330‒332, Line 339, and Line 342: Proximal and distal are adjectives related to “distance”, rather than “direction”.

Pg. 17 Lines 346‒353: Also discuss if there was any difference in performance and/or learning between external and holistic foci.

Pg. 17 Lines 354‒355: The meaning of this sentence is unclear. Elaborate or rephrase for clarity.

Pg. 17 Lines 365‒367: Provide an explanation for the decision to omit the grey literature in this review.

Pg. 18 Lines 373‒375: Elaborate on why these recommendations were made for future studies. For example, why all, rather than specific, skill levels, and why these two particular age groups since they are represented in the nine included studies with participants ranging from eight to 33 years of age.

Major Comments – Conclusion

Pg. 18 Line 381: For clarity, provide more than one example of these “alternatives”.

Minor Comments

Pg. 2 Line 37: Consider replacing “by” with “using”, “following”, “according to”, or similar.

Pg. 2 Line 42: Replace slashes with “or” throughout the manuscript (e.g., Pg. 2 Line 42, Pg. 3 Line 58).

Pg. 2 Lines 43‒44: Use parentheses as in “(e.g., different…instruction)”, or spell out the abbreviation.

Pg. 2 Line 44: Replace with “task(s)” and “skill(s)”.

Pg. 2 Line 54: Check expression of the phrase structure in “on a high level of advance requires the player”.

Pg. 3 Lines 63‒65: Check usage of “connection” and sentence structure.

Pg. 3 Line 78: Delete “however”.

Pg. 4 Line 81: As a verb, “agree” is not usually used with the noun “research”.

Pg. 4 Line 84: Add “relative” before “to”.

Pg. 4 Line 91: Do you mean “aspects close to the body”?

Pg. 4 Line 95: Add “of” before “attention”.

Pg. 5 Lines 115‒116: Use parentheses as in “(e.g., conference abstracts)”, or spell out the abbreviation. Likewise for Pg. 5 Line 122.

Pg. 5 Lines 127‒128: Check sentence structure.

Pg. 6 Lines 143‒144: There is ambiguity in this sentence in terms of how each category was applied to each included study.

Pg. 7 Lines 153‒154: Check sentence structure and consider using “respectively” to indicate that the numerical values are associated with the terms in the given order, rather than refer to a numerical range.

Pg. 8 Line 183: Check sentence structure.

Pg. 8 Line 187: “Participant characteristics”

Pg. 8 Line 188: The use of “particular” seems to imply that this sample size range applied to only a subset of the nine included studies.

Pg. 8 Line 189: Delete “The”.

Pg. 8 Line 192: Do you mean “skill” (rather than “sport”) level?

Pg. 8 Line 193: Check sentence structure of “Three…size”.

Pg. 8 Lines 193 and 195, and Pg. 9 Lines 198 and 211: There is a mismatch between the use of “a” and plural nouns.

Pg. 8 Line 194: Add a comma after “[31,32]”.

Pg. 9 Line 198: The verb “perform” is not commonly used with “design” as the noun.

Pg. 9 Lines 202‒203, Lines 218‒219, and Lines 220‒221, and Pg. 10 Lines 222‒223, and Line 224: It is unusual to have only one sentence in a paragraph.

Pg. 9 Line 204: What is “tennis table”?

Pg. 9 Line 205: “examining”

Pg. 9 Line 207: Add a comma after “[30]”.

Pg. 9 Line 210: Delete “of” before “experimental”.

Pg. 9 Line 216: “overview” is a noun, but used as a verb here.

Pg. 9 Line 221: Add “and” before “accuracy”.

Pg. 12 Lines 237‒239: Check sentence structure.

Pg. 13 Line 242: Add “and” before “with”.

Pg. 13 Line 251: Replace “other” with “another”.

Pg. 13 Line 257: Do you mean “from the pretest to the post-test”?

Pg. 13 Lines 255‒256: It is confusing when both “without” and “with” are used in the same sentence.

Pg. 13 Lines 261‒263: Check sentence structure.

Pg. 13 Lines 265‒266: Check sentence structure. It is also not clear what “included research” refers to.

Pg. 14 Line 271: Awkward phrase in “between an external and IF group”. Use either the full form or abbreviation, but not both, when describing different attentional focus conditions.

Pg. 14 Line 272: Use either the full form or abbreviation, but not both, when describing different attentional focus conditions.

Pg. 14 Line 273: Are you referring to the “participants” rather than the “authors”?

Pg. 14 Lines 276‒277: Incorrect matching of singular and plural words.

Pg. 14 Line 284: Mismatch between the use of “a” and a plural noun.

Pg. 14 Line 285: Rephrase as “an IF or control condition”.

Pg. 14 Line 289: Replace “of the” with “on”.

Pg. 14 Line 291 to Pg. 15 Line 296: Check structure of these three sentences.

Pg. 15 Line 302: Delete “tasks”.

Pg. 15 Line 318: Add “those” before “described”.

Pg. 16 Line 319: Add “which” before “surprisingly”.

Pg. 16 Line 322: instruction’s

Pg. 16 Line 324: Replace “subjects’ ” with “participant’s”.

Pg. 16 Lines 324‒325: Use parentheses as in “(e.g., “place…grip”)”, or spell out the abbreviation.

Pg. 16 Line 325: Replace “hold” with “held”, and add “the” before “racket itself”.

Pg. 16 Line 326: Delete “a”.

Pg. 16 Line 327: Delete “and”.

Pg. 16 Line 328: participant’s

Pg. 16 Line 329: Use parentheses as in “(e.g., “concentrate…table”)”, or spell out the abbreviation.

Pg. 16 Line 335: Delete “a” before “distal”.

Pg. 16 Line 338: “target”

Pg. 16 Line 340: Replace “contradict” with “in contrast”, and add “may” after “who”.

Pg. 16 Line 341: Use “Given that”. The adjectives “novice” and “low skilled” refer to the sample, rather than its size.

Pg. 16 Line 342: Replace with “participants was used, this…” and consider replacing “assumption” with “explanation”.

Pg. 16 Line 343: Use “appear” and delete “a”.

Pg. 17 Line 361: Check the proper use of ellipsis.

Pg. 17 Line 363: Add a character space before “[26]”.

Pg. 18 Line 374: Add “for” before “all”. What does “young” refer to?

Pg. 18 Line 380: It is inappropriate to use “concludes” in this context.

Pg. 21 Figure 1: Insert spaces before and after the equal signs. Do not introduce new terms in the figures and tables; the same terms as those used in the main text should be used. That is, what does “reference group” mean?

Reviewer #2: This paper concerns a review of the literature on the effects of attentional focus on the performance/learning of racket sports. A descriptive synthesis is presented of nine studies. Overall, the review concludes that external focus seems to promote performance and learning in this context, especially when compared to an internal focus intervention.

Overall, the review contains all the elements required for a systematic review, and will be of use for a particular audience interested in this specific area of research. However, I do believe that there is room for improvement, and have listed my main points of feedback below.

First, regarding the background of the review, there is a need to better explain why this review is relevant. That is, given other recent research in this area, including the recent review by Chua et al cited by the authors, why would it be useful to do a review focused on racket sports in particular? Do these type

Of sports have particular characteristics that set them apart from other sports (ie biomechanically, or in terms of attentional demands) which warrants a separate review on these types of skills/sports? It would be good if this can be further explained.

I also think that the introduction and discussion can be improved by incorporating recent papers that cast doubt on the (strength of the) attentional focus effects. That is, many authors have proposed that the attentional focus effects may not be as strong and unequivocal as previously thought, and that (i) publication bias may have distorted the current evidence base (https://sportrxiv.org/index.php/server/preprint/view/304), and also that there are good theoretical grounds to assume that internal attentional focus may promote performance and learning in certain conditions (https://www.frontiersin.org/articles/10.3389/fspor.2023.1176635/full). I would propose that the current review could be strengthened by integrating some of these recent advances in our understanding of the field, as it would provide for a more balanced discussion of the evidence. This would apply both to the introduction and discussion sections.

The systematic search was registered, which is good. The eligibility criteria were a bit vague, however. Could the authors specify the exact designs they were interested in (RCTs, non randomised studies, other?) and also the specific requirements regarding performance vs learning assessment? Ie online effects (immediate effects on performance) may differ from learning effects, measures with retention tests. It seems both types of studies were eligible though? Please clarify.

Regarding the Pedro assessment: it is good that some assessment of study quality was performed, but I don’t feel this has really been factored into the synthesis / interpretation of the evidence. Did studies with lower risk of bias show different effects to studies with higher risk of bias? At the very least, the discussion section should feature a discussion of the methodological issues noted, and how these should affect our interpretation/ weighting of the strength of the evidence.

The descriptive synthesis (sometimes referred to as “qualitative” synthesis, but that term should not be used here) is not really described within the methods section. Could the authors please insert a paragraph to explain their descriptive synthesis approach? This could then also include a description of the reasoning for not doing a meta analysis (which I agree was not suitable).

Reason I am asking is also that the descriptive synthesis was not “synthesising” results sufficiently in my opinion. Ie there was room to structure the results better, to help guide the reader to the findings. eg first present results of external vs internal focus comparisons, then for external vs control comparisons, and first do this for studies with immediate effects followed by presenting results for studies with delayed retention tests. An overview table simply showing for each study of results showed significant differences consistent with significantly superior outcomes for external focus, no effect, or better outcomes for internal focus could be very helpful too (see eg my own review on implicit learning for an example of how this can be done: https://journals.plos.org/plosone/article?id=10.1371/journal.pone.0203591).

I think a key issue in this literature is that many studies, including some of my own, have failed to test if people indeed focused as instructed (and if they, say, reported less mental effort or focus toward movement when focusing externally). Has this been checked at all in the included studies? If not, is there a risk that any effects found may be due to other differences between the external and internal focus interventions?

Holistic focus and analogy instructions are brought up as an alternative to an external focus - or at least that’s how I understood it from the paper. Could the authors please clarify the conceptual overlap between these two and external focus? Can’t both holistic and analogy instructions lead to internally focused attention? I.e. we once gave children the instruction to move their arms like a pendulum when putting, which arguably directs attention to movement. Also, one example of an holistic instruction given in the review is very internally focused indeed, emphasising the “feel” of the movement. Please explain how we should interpret these interventions in the context of the external vs internal

Focus literature.

6. PLOS authors have the option to publish the peer review history of their article (what does this mean?). If published, this will include your full peer review and any attached files.

Reviewer #1: No

Reviewer #2: **Yes: **Elmar Kal

---

## [Author Response · Author response to Decision Letter 0]

6 Nov 2023

RESPONSE TO REVIEWERS COMMENTS

Dear Editor and Reviewers,

The authors would like to thank the reviewers for their precious time and invaluable comments. We have carefully addressed all the comments. The corresponding changes and refinements made in the revised paper are summarized in our response below. We hope that the modifications and explanations will be acceptable for you.

Reviewer #1

Major Comments – Abstract

1. Pg. 2 Line 45: Does “paddle” refer to “padel” or “paddle tennis”? If it refers to the former, use “padel” to avoid ambiguity because it is the more popularly used term, and to be consistent with its usage elsewhere.

Response: We use the term “padel” to refer to both “paddle tennis” and “paddle” in the revised manuscript to ensure clarity.

Major Comments – Method

Pg. 5 Line 117: Figure 1 mentioned “other sources”, but the search for relevant articles other than the use of the four electronic databases has not been described under this subsection on search strategy.

Response: Including this section in the flow chart diagram might lead to confusion since we have not collected data from additional sources. This has been changed.

Pg. 5 Line 124: See comment on abstract regarding “paddle”.

Response: This has been changed.

Pg. 5 Line 128: How was the de-duplication process “automatic”?

Response: We have rephrased this sentence for clarity.

Pg. 6 Line 133: Clarify “level”.

Response: We have rephrased this sentence for clarity.

Pg. 6 Line 140: Since no scoring system has been described, it is unclear what “total score” means.

Response: We have made changes and added more detailed information.

Pg. 6 Line 142: Who does “they” refer to?

Response: It refers to subjects and therapist: “…it is not feasible to blind the subjects and therapists (items 5 and 6), as they are actively participating in the exercise program.”

Pg. 6 Lines 145‒146: The kappa coefficient is a statistic and is obtained using a type of correlation statistics known as the interrater reliability test. This test assesses the level of agreement amongst raters; it is not able to influence (control for) risk of bias.

Response: We have rephrased this sentence for clarity.

Major Comments – Results

Pg. 7 Line 153 and 159: It is unclear if the same type of agreement analysis was conducted to evaluate the interrater reliability for abstract screening, for full-text screening, and for risk of bias assessment, or whether the same number of raters (reviewers) were used in the assessment, because either absolute or mean (rather than the same type of) kappa coefficient values were reported.

Response: The agreement between reviewers assessing the quality of studies and title and abstract screening processes was verified using a Cohen’s kappa correlation. The number of raters who conducted the assessment is shown in parentheses throughout the text. To enhance clarity, we have included only the absolute kappa coefficient values in the revised manuscript. 

Pg. 8 Line 180: See comment on abstract regarding “paddle”.

Response: This has been changed.

Pg. 8 Line 188: Does “452” refer to the total number of participants combined across the nine included studies? There is no need to specify “men and women” because it would be clear from the subsequent sentences that the participants consisted of both males and females in the children and adult age groups.

Response: We have removed the gender specific term from this sentence.

Pg. 8 Line 191: Why would there be seniors as young as 33 years of age? Remove the comma.

Response: This has been changed.

Pg. 8 Lines 195‒196: Clarify the context for “practice experience”. How is it different from “training experience” or simply “experience”?

Response: Both terms are used by the authors to describe a period of involvement in a structured, organized training and competition. We have retained the term “experience” in the revised paper.

Pg. 9 Line 200: For readers who are not so familiar with the motor learning literature, an elaboration on the “immediate performance effect” may be required.

Response: Thank you for this suggestion. We are unsure if the Results section is an appropriate place to discuss this issue. However, we have revised this sentence to improve its clarity.

Pg. 9 Lines 207‒208: There is conceptual incongruence between “direction” and “proximal, distal or increasing in distance”. That is, it is unclear why this list was provided in parentheses as examples for “direction”.

Response: Attentional focus can be viewed from different perspectives and has been characterized, for example, in terms of its direction. Moreover, an external focus of attention may be directed close to the body (proximal) or farther away from the body (distal). For further details, see, for example, Wulf et al. (2015). The list of instructions provided within the parentheses pertains to the specific instructions outlined by the authors of the cited papers in this sentence. 

Pg. 10 Lines 222‒223: Were these improvements found at a particular assessment timepoint (e.g., post-test, transfer test) or multiple timepoints?

Response: We have added accurate information to this sentence.

Pg. 10 Line 228: What is “base”?

Response: According to the authors, base is considered one of the key aspects that contribute to the proper game performance in tennis. The term refers to the appropriate return of a performer to a “home” or “recovery” position in between skill attempts.

Pg. 10 Lines 229‒230: Provide the contextual clarification for “directed externally”. The comparative use of “participants” versus “athletes” in the same sentence seems to imply that there could be a mixture of non-athletes and athletes in the sample, and that they were not randomly assigned to the various treatment groups.

Response: This has been changed.

Pg. 10 Line 231: Does “base evaluation” refer to the baseline assessment administered before the start of the acquisition (practice) phase?

Response: The term “base” refers to the term described in one of the previous comment. To enhance clarity, we have made a revision to the manuscript by capitalizing the term “base” at the beginning. 

Pg. 13 Lines 256‒258: Rephrase so that it is clear that motor performance improvements did not differ between the distal EF and proximal EF groups, but both groups performed better than the IF group, for both the post-test and retention test.

Response: This has been changed.

Pg. 13 Lines 258‒260: What is the “second research”? The two “external instructions” should be described in a way that shows the differentiation between them.

Response: This has been changed.

Pg. 14 Line 269: Were these “findings” from Abedanzadeh et al. or the three studies referenced in the first sentence of this paragraph?

Response: This sentence refers to the research conducted by Abedanzadeh et al. This sentence has been moved to a new paragraph to improve readability throughout the text.

Pg. 273 Line 273: Retention and transfer tests are typically administered with a delay of at least 24 hours after the last practice trial. Does “immediately” mean these tests occurred within the same day as the last practice session?

Response: We agree that these tests are typically conducted with a time delay, often within 24 hours or even less. However, based on the methodology section of the study, we can only conclude that the retention and transfer tests were carried out consecutively, immediately following the final training block. No additional details were provided regarding the time intervals between each test.

Pg. 14 Line 279: It is unclear which condition(s) the combined effects of augmented feedback and EF were not better than.

Response: This has been changed.

Major Comments – Discussion

Pg. 14 Lines 281‒282: According to the review aim described on Pg. 3, both learning and performance effects were of interest.

Response: This has been changed.

Pg. 16 Lines 330‒332, Line 339, and Line 342: Proximal and distal are adjectives related to “distance”, rather than “direction”.

Response: We have already discussed this issue in one of our previous comments.

Pg. 17 Lines 346‒353: Also discuss if there was any difference in performance and/or learning between external and holistic foci.

Response: We have addressed this issue in the revised manuscript.

Pg. 17 Lines 354‒355: The meaning of this sentence is unclear. Elaborate or rephrase for clarity.

Response: We have rephrased this sentences. 

Pg. 17 Lines 365‒367: Provide an explanation for the decision to omit the grey literature in this review.

Response: We firmly believe that the grey literature holds immense potential to offer valuable and pertinent resources during the process of evidence synthesis. However, throughout the process of preparing a systematic review, we made the deliberate decision to only include peer-reviewed literature. This selection ensures that the articles we include are of the highest quality possible. We have made the assumption that this selection process guarantees that the articles we choose are of the utmost highest quality.

Pg. 18 Lines 373‒375: Elaborate on why these recommendations were made for future studies. For example, why all, rather than specific, skill levels, and why these two particular age groups since they are represented in the nine included studies with participants ranging from eight to 33 years of age.

Response: The recent studies conducted by McKay et al. (2023) and Gottwald et al. (2023) have brought forth valuable insights that hold great promise in broadening our understanding of attentional focus. These studies, from both methodological and theoretical standpoints, offer significant contributions that are sure to shape the future of this field. Firstly, McKay et al. (2023) challenge the existing evidence supporting the superiority of external focus of attention. By highlighting methodological biases that have thus far been overlooked, their findings call for a reassessment of future research in order to obtain a more accurate understanding. Furthermore, while the ecological dynamics approach provides a strong conceptual framework for attentional focus, there remains a noticeable dearth of empirical research and quantitative data to support its claims. This underscores the need for further investigation in this area, allowing for a solid scientific basis to be established. Additionally, considering the scarcity of studies conducted on attentional focus in racket sports, research must continue in order to better comprehend the specific context. This applies to individuals of all levels of expertise and genders, as the findings can provide valuable insights that can enhance performance and optimize training methods.

Major Comments – Conclusion

Pg. 18 Line 381: For clarity, provide more than one example of these “alternatives”.

Response: We have added accurate information to this sentence.

Minor Comments

Pg. 2 Line 37: Consider replacing “by” with “using”, “following”, “according to”, or similar.

Response: This has been changed.

Pg. 2 Line 42: Replace slashes with “or” throughout the manuscript (e.g., Pg. 2 Line 42, Pg. 3 Line 58).

Response: This has been changed.

Pg. 2 Lines 43‒44: Use parentheses as in “(e.g., different…instruction)”, or spell out the abbreviation.

Response: This has been changed.

Pg. 2 Line 44: Replace with “task(s)” and “skill(s)”.

Response: This has been changed.

Pg. 2 Line 54: Check expression of the phrase structure in “on a high level of advance requires the player”.

Response: We have rephrased this paragraph.

Pg. 3 Lines 63‒65: Check usage of “connection” and sentence structure.

Response: We have rephrased this paragraph.

Pg. 3 Line 78: Delete “however”.

Response: We have rephrased this sentence.

Pg. 4 Line 81: As a verb, “agree” is not usually used with the noun “research”.

Response: We have rephrased this sentence.

Pg. 4 Line 84: Add “relative” before “to”.

Response: This has been changed.

Pg. 4 Line 91: Do you mean “aspects close to the body”?

Response: This has been changed.

Pg. 4 Line 95: Add “of” before “attention”.

Response: This has been changed.

Pg. 5 Lines 115‒116: Use parentheses as in “(e.g., conference abstracts)”, or spell out the abbreviation. Likewise for Pg. 5 Line 122.

Response: This has been changed.

Pg. 5 Lines 127‒128: Check sentence structure.

Response: We have rephrased this sentence.

Pg. 6 Lines 143‒144: There is ambiguity in this sentence in terms of how each category was applied to each included study.

Response: We have revised the wording of this sentence. 

Pg. 7 Lines 153‒154: Check sentence structure and consider using “respectively” to indicate that the numerical values are associated with the terms in the given order, rather than refer to a numerical range.

Response: This has been changed.

Pg. 8 Line 183: Check sentence structure.

Response: We have rephrased this sentence.

Pg. 8 Line 187: “Participant characteristics”

Response: This has been changed.

Pg. 8 Line 188: The use of “particular” seems to imply that this sample size range applied to only a subset of the nine included studies.

Response: This has been changed.

Pg. 8 Line 189: Delete “The”.

Response: This has been changed.

Pg. 8 Line 192: Do you mean “skill” (rather than “sport”) level?

Response: This has been changed.

Pg. 8 Line 193: Check sentence structure of “Three…size”.

Response: We have rephrased this sentence.

Pg. 8 Lines 193 and 195, and Pg. 9 Lines 198 and 211: There is a mismatch between the use of “a” and plural nouns.

Response: We have rephrased this sentence.

Pg. 8 Line 194: Add a comma after “[31,32]”.

Response: This has been changed.

Pg. 9 Line 198: The verb “perform” is not commonly used with “design” as the noun.

Response: We have rephrased this sentence.

Pg. 9 Lines 202‒203, Lines 218‒219, and Lines 220‒221, and Pg. 10 Lines 222‒223, and Line 224: It is unusual to have only one sentence in a paragraph.

Response: This has been changed.

Pg. 9 Line 204: What is “tennis table”?

Response: This has been changed.

Pg. 9 Line 205: “examining”

Response: This has been changed.

Pg. 9 Line 207: Add a comma after “[30]”.

Response: This has been changed.

Pg. 9 Line 210: Delete “of” before “experimental”.

Response: This has been changed.

Pg. 9 Line 216: “overview” is a noun, but used as a verb here.

Response: This has been changed.

Pg. 9 Line 221: Add “and” before “accuracy”.

Response: This has been changed.

Pg. 12 Lines 237‒239: Check sentence structure.

Response: We have rephrased this sentence.

Pg. 13 Line 242: Add “and” before “with”.

Response: This has been changed.

Pg. 13 Line 251: Replace “other” with “another”.

Response: This has been changed.

Pg. 13 Line 257: Do you mean “from the pretest to the post-test”?

Response: We have rephrased this sentence.

Pg. 13 Lines 255‒256: It is confusing when both “without” and “with” are used in the same sentence.

Response: We have rephrased this sentence.

Pg. 13 Lines 261‒263: Check sentence structure.

Response: We have rephrased this sentence.

Pg. 13 Lines 265‒266: Check sentence structure. It is also not clear what “included research” refers to.

Response: We have rephrased this sentence.

Pg. 14 Line 271: Awkward phrase in “between an external and IF group”. Use either the full form or abbreviation, but not both, when describing different attentional focus conditions.

Response: This has been changed.

Pg. 14 Line 272: Use either the full form or abbreviation, but not both, when describing different attentional focus conditions.

Response: This has been changed.

Pg. 14 Line 273: Are you referring to the “participants” rather than the “authors”?

Response: We have rephrased this sentence.

Pg. 14 Lines 276‒277: Incorrect matching of singular and plural words.

Response: We have rephrased this sentence.

Pg. 14 Line 284: Mismatch between the use of “a” and a plural noun.

Response: This has been changed.

Pg. 14 Line 285: Rephrase as “an IF or control condition”.

Response: This has been changed.

Pg. 14 Line 289: Replace “of the” with “on”.

Response: This has been changed.

Pg. 14 Line 291 to Pg. 15 Line 296: Check structure of these three sentences.

Response: We have rephrased this sentence.

Pg. 15 Line 302: Delete “tasks”.

Response: This has been changed.

Pg. 15 Line 318: Add “those” before “described”.

Response: This has been changed.

Pg. 16 Line 319: Add “which” before “surprisingly”.

Response: This has been changed.

Pg. 16 Line 322: instruction’s

Response: This has been changed.

Pg. 16 Line 324: Replace “subjects’ ” with “participant’s”.

Response: This has been changed.

Pg. 16 Lines 324‒325: Use parentheses as in “(e.g., “place…grip”)”, or spell out the abbreviation.

Response: This has been changed.

Pg. 16 Line 325: Replace “hold” with “held”, and add “the” before “racket itself”.

Response: This has been changed.

Pg. 16 Line 326: Delete “a”.

Response: This has been changed.

Pg. 16 Line 327: Delete “and”.

Response: This has been changed.

Pg. 16 Line 328: participant’s

Response: This has been changed.

Pg. 16 Line 329: Use parentheses as in “(e.g., “concentrate…table”)”, or spell out the abbreviation.

Response: This has been changed.

Pg. 16 Line 335: Delete “a” before “distal”.

Response: This has been changed.

Pg. 16 Line 338: “target”

Response: This has been changed.

Pg. 16 Line 340: Replace “contradict” with “in contrast”, and add “may” after “who”.

Response: This has been changed.

Pg. 16 Line 341: Use “Given that”. The adjectives “novice” and “low skilled” refer to the sample, rather than its size.

Response: We have rephrased this sentence.

Pg. 16 Line 342: Replace with “participants was used, this…” and consider replacing “assumption” with “explanation”.

Response: This has been changed.

Pg. 16 Line 343: Use “appear” and delete “a”.

Response: This has been changed.

Pg. 17 Line 361: Check the proper use of ellipsis.

Response: This has been changed.

Pg. 17 Line 363: Add a character space before “[26]”.

Response: This has been changed.

Pg. 18 Line 374: Add “for” before “all”. What does “young” refer to?

Response: This has been changed.

Pg. 18 Line 380: It is inappropriate to use “concludes” in this context.

Response: This has been changed.

Pg. 21 Figure 1: Insert spaces before and after the equal signs. Do not introduce new terms in the figures and tables; the same terms as those used in the main text should be used. That is, what does “reference group” mean?

Response: This has been changed.

Reviewer #2

This paper concerns a review of the literature on the effects of attentional focus on the performance/learning of racket sports. A descriptive synthesis is presented of nine studies. Overall, the review concludes that external focus seems to promote performance and learning in this context, especially when compared to an internal focus intervention.

Overall, the review contains all the elements required for a systematic review, and will be of use for a particular audience interested in this specific area of research. However, I do believe that there is room for improvement, and have listed my main points of feedback below.

First, regarding the background of the review, there is a need to better explain why this review is relevant. That is, given other recent research in this area, including the recent review by Chua et al cited by the authors, why would it be useful to do a review focused on racket sports in particular? Do these type

Of sports have particular characteristics that set them apart from other sports (ie biomechanically, or in terms of attentional demands) which warrants a separate review on these types of skills/sports? It would be good if this can be further explained.

Response: We have addressed these issues in the revised manuscript.

I also think that the introduction and discussion can be improved by incorporating recent papers that cast doubt on the (strength of the) attentional focus effects. That is, many authors have proposed that the attentional focus effects may not be as strong and unequivocal as previously thought, and that (i) publication bias may have distorted the current evidence base (https://sportrxiv.org/index.php/server/preprint/view/304), and also that there are good theoretical grounds to assume that internal attentional focus may promote performance and learning in certain conditions (https://www.frontiersin.org/articles/10.3389/fspor.2023.1176635/full). I would propose that the current review could be strengthened by integrating some of these recent advances in our understanding of the field, as it would provide for a more balanced discussion of the evidence. This would apply both to the introduction and discussion sections.

Response: Thank you for your valuable suggestions. According to your comments, we have included some insights from recent research findings into both our introduction and discussion sections of the revised manuscript. Both studies have immense value from both methodological and theoretical perspectives, and undoubtedly have the potential to significantly enhance our comprehension of attentional focus in the future. Although the ecological dynamics approach provides a solid conceptual framework, there is still a notable lack of empirical research and quantitative data to support its claims in the field of attentional focus.

The systematic search was registered, which is good. The eligibility criteria were a bit vague, however. Could the authors specify the exact designs they were interested in (RCTs, non randomised studies, other?) and also the specific requirements regarding performance vs learning assessment? Ie online effects (immediate effects on performance) may differ from learning effects, measures with retention tests. It seems both types of studies were eligible though? Please clarify.

Response: During the formulation of the research question and objectives for our systematic review, we consulted experts on the topic of the review and utilized our own scientific experiences on attentional focus in sports. Despite cognitive aspects being crucial to racket sports performance, we noticed that the literature only briefly discusses the effectiveness of attentional focus instructions. As a result, we decided to conduct a more comprehensive search of the available evidence, avoiding limitations such as specific learning and motor performance requirements or randomized controlled trials, to avoid losing potentially relevant records.

Regarding the Pedro assessment: it is good that some assessment of study quality was performed, but I don’t feel this has really been factored into the synthesis / interpretation of the evidence. Did studies with lower risk of bias show different effects to studies with higher risk of bias? At the very least, the discussion section should feature a discussion of the methodological issues noted, and how these should affect our interpretation/ weighting of the strength of the evidence.

Response: We have included a new paragraph in the revised manuscript that discusses the methodological quality of the studies.

The descriptive synthesis (sometimes referred to as “qualitative” synthesis, but that term should not be used here) is not really described within the methods section. Could the authors please insert a paragraph to explain their descriptive synthesis approach? This could then also include a description of the reasoning for not doing a meta analysis (which I agree was not suitable).

Response: We have outlined the reasons for not conducting a meta-analysis in the limitations section of our study.

Reason I am asking is also that the descriptive synthesis was not “synthesising” results sufficiently in my opinion. Ie there was room to structure the results better, to help guide the reader to the findings. eg first present results of external vs internal focus comparisons, then for external vs control comparisons, and first do this for studies with immediate effects followed by presenting results for studies with delayed retention tests. An overview table simply showing for each study of results showed significant differences consistent with significantly superior outcomes for external focus, no effect, or better outcomes for internal focus could be very helpful too (see eg my own review on implicit learning for an example of how this can be done: https://journals.plos.org/plosone/article?id=10.1371/journal.pone.0203591).

Response: Thank you for your comments and suggestions. We presented the results of the EF versus IF and control comparison, then we presented the results of alternative focus strategies, both for performance and learning effects. Considering the weakness of the methodological quality and diversity of extracted studies, we decided to present a more brief and narrative synthesis of the findings in this systematic review. In the present version, we have made substantial changes in several parts of the results and discussion sections.

I think a key issue in this literature is that many studies, including some of my own, have failed to test if people indeed focused as instructed (and if they, say, reported less mental effort or focus toward movement when focusing externally). Has this been checked at all in the included studies? If not, is there a risk that any effects found may be due to other differences between the external and internal focus interventions?

Response: We have addressed this issue in the revised manuscript. Furthermore, we have included additional information in Table 1 concerning the implementation of the manipulation check procedure in the included studies.

Holistic focus and analogy instructions are brought up as an alternative to an external focus - or at least that’s how I understood it from the paper. Could the authors please clarify the conceptual overlap between these two and external focus? Can’t both holistic and analogy instructions lead to internally focused attention? I.e. we once gave children the instruction to move their arms like a pendulum when putting, which arguably directs attention to movement. Also, one example of an holistic instruction given in the review is very internally focused indeed, emphasising the “feel” of the movement. Please explain how we should interpret these interventions in the context of the external vs internal Focus literature.

Response: We have made substantial changes in the paragraph to address the editors’ comments.

---

## [Decision Letter · Decision Letter 1]

28 Nov 2023

PONE-D-23-11299R1Attentional focus strategies in racket sports: A systematic reviewPLOS ONE

Dear Dr. Niźnikowski,

Thank you for submitting your manuscript to PLOS ONE. After careful consideration, we feel that it has merit but does not fully meet PLOS ONE’s publication criteria as it currently stands. Therefore, we invite you to submit a revised version of the manuscript that addresses the points raised during the review process.

We look forward to receiving your revised manuscript.

Kind regards,

Nick Fogt

Academic Editor

PLOS ONE

Journal Requirements:

Additional Editor Comments (if provided):

Both reviewers agree that the paper is improved. However, there are some additional comments from the reviewers that will need to be addressed. Please address each comment in detail.

Reviewers' comments:

Reviewer's Responses to Questions

**Comments to the Author**

1. If the authors have adequately addressed your comments raised in a previous round of review and you feel that this manuscript is now acceptable for publication, you may indicate that here to bypass the “Comments to the Author” section, enter your conflict of interest statement in the “Confidential to Editor” section, and submit your "Accept" recommendation.

Reviewer #1: (No Response)

Reviewer #2: (No Response)

2. Is the manuscript technically sound, and do the data support the conclusions?

Reviewer #1: Yes

Reviewer #2: Yes

3. Has the statistical analysis been performed appropriately and rigorously? 

Reviewer #1: Yes

Reviewer #2: N/A

4. Have the authors made all data underlying the findings in their manuscript fully available?

Reviewer #1: Yes

Reviewer #2: No

5. Is the manuscript presented in an intelligible fashion and written in standard English?

Reviewer #1: Yes

Reviewer #2: Yes

6. Review Comments to the Author

Reviewer #1: Minor Comments

1. Pg. 3 Lines 66-67: Is the opponent’s side of the table also referred to as “the opponent’s court” in the racket sport of table tennis?

2. Pg. 7 Line 162: Do you mean “≥6”?

3. Pg. 9 Line 212: Check the expression in “Three of the nine…their sample size” so that its intended meaning is clear.

4. Pg. 9 Line 214: Remove the comma after the brackets.

5. Pg. 11 Table 2: Check the consistency in using upper- and/or lower-case letters for the column headings.

6. Pg. 13 Table Footnote: Check the consistency in using hyphens or em dashes.

Reviewer #2: I would like to thank the authors for addressing most of my comments. For future submissions, I would like to advise the authors to include textual changes made as part of their response, as that will make it easier to review (and avoids any potential misinterpretations as part of this process).

I have a few additional suggestions/comments, especially with regard to the discussion and conclusion section:

- Discussion - page 16, lines 320-321: “It has been noted that the advantages of external focus bring about a variety of effects that we cannot fully comprehend. On average, however, these effects tend to be minimal or even non-existent [42].”

It is not clear from this sentence what the advantages are that we cannot fully comprehend or why they appear to be minimal or non-existent. Either clarify this further or consider removing altogether.

- Discussion – page 16, lines 322-327: It is stated that the findings presented here support the predictions of the constrained action hypothesis, but that seems to be overinterpreting the current results. That is, the constrained action hypothesis specifically predicts that movement performance is enhanced through increased automaticity, but based on the results presented it seems impossible to make any conclusions as to whether those that received an external focus indeed moved more automatically (this would require assessments of outcomes indicative of automaticity, which are not really presented here).

- Discussion – page 17,18, lines 367-373: I like the additional discussion on the potential benefit of proximal EF for novices here. The final conclusion presented here regarding skill differences and proximal/distal EF seems to be inconsistent with the information presented before though: Based on the findings in the review, is the main conclusion that skill level does not make a difference (and that both proximal and distal EF would be equally effective)? If so then please clarify why this is concluded, as I wonder if the evidence presented in the review sufficiently strong to make this claim?

- I think a key limitation of the reviewed studies (as with most studies in motor learning) is the limited follow-up period. As skill acquisition in practice typically unfolds at far longer time scales, we have to acknowledge that there is inherently limited scope to generalise results from studies like the ones reviewed here to real-world application. It would be good to discuss this explicitly in the review’s discussion.

- Given the limited evidence available for the use of holistic cues or similar, I would remove reference to these / the recommendation to explore these from the main conclusion section within the discussion (and just discuss this only as part of the rest of the main text, as potential avenue for research and application).

7. PLOS authors have the option to publish the peer review history of their article (what does this mean?). If published, this will include your full peer review and any attached files.

Reviewer #1: No

Reviewer #2: No

---

## [Author Response · Author response to Decision Letter 1]

8 Dec 2023

RESPONSE TO REVIEWERS COMMENTS

Dear Editor and Reviewers,

The authors would like to thank the Reviewers for the opportunity to improve our manuscript for the second time. All requested changes have been included in the revised paper and highlighted in yellow. Likewise, we hope that our modifications and explanations meet your expectations. We feel our revised manuscript is much improved.

Reviewer #1

Minor Comments

1. Pg. 3 Lines 66-67: Is the opponent’s side of the table also referred to as “the opponent’s court” in the racket sport of table tennis?

Response: We have rephrased this sentence for clarity.

2. Pg. 7 Line 162: Do you mean “≥6”?

Response: This has been changed.

3. Pg. 9 Line 212: Check the expression in “Three of the nine…their sample size” so that its intended meaning is clear.

Response: We have rephrased this sentence for clarity.

4. Pg. 9 Line 214: Remove the comma after the brackets.

Response: This has been changed.

5. Pg. 11 Table 2: Check the consistency in using upper- and/or lower-case letters for the column headings.

Response: This has been changed.

6. Pg. 13 Table Footnote: Check the consistency in using hyphens or em dashes.

Response: This has been changed.

Reviewer #2

I would like to thank the authors for addressing most of my comments. For future submissions, I would like to advise the authors to include textual changes made as part of their response, as that will make it easier to review (and avoids any potential misinterpretations as part of this process).

Response: We would like to thank the Reviewer for the thoughtful comments that were provided on the previous version of this paper. We have revised the requested sections according to the above suggestions. Please see the highlighted and indicated text in the lines. 

I have a few additional suggestions/comments, especially with regard to the discussion and conclusion section:

- Discussion - page 16, lines 320-321: “It has been noted that the advantages of external focus bring about a variety of effects that we cannot fully comprehend. On average, however, these effects tend to be minimal or even non-existent [42].”

It is not clear from this sentence what the advantages are that we cannot fully comprehend or why they appear to be minimal or non-existent. Either clarify this further or consider removing altogether.

Response: We have rephrased these sentences for better clarity, as shown in lines 321–328.

- Discussion – page 16, lines 322-327: It is stated that the findings presented here support the predictions of the constrained action hypothesis, but that seems to be overinterpreting the current results. That is, the constrained action hypothesis specifically predicts that movement performance is enhanced through increased automaticity, but based on the results presented it seems impossible to make any conclusions as to whether those that received an external focus indeed moved more automatically (this would require assessments of outcomes indicative of automaticity, which are not really presented here).

Response: We agree with the comment, and we have addressed the concern mentioned by removing the statements on the constrained action hypothesis and have rephrased the paragraph to improve clarity and understanding, as presented in lines 329–341.

- Discussion – page 17,18, lines 367-373: I like the additional discussion on the potential benefit of proximal EF for novices here. The final conclusion presented here regarding skill differences and proximal/distal EF seems to be inconsistent with the information presented before though: Based on the findings in the review, is the main conclusion that skill level does not make a difference (and that both proximal and distal EF would be equally effective)? If so then please clarify why this is concluded, as I wonder if the evidence presented in the review sufficiently strong to make this claim?

Response: We agree with the reviewer, and we have revised the paragraph to address the inconsistency noted by the reviewer, as shown in lines 371–388. 

- I think a key limitation of the reviewed studies (as with most studies in motor learning) is the limited follow-up period. As skill acquisition in practice typically unfolds at far longer time scales, we have to acknowledge that there is inherently limited scope to generalise results from studies like the ones reviewed here to real-world application. It would be good to discuss this explicitly in the review’s discussion.

Response: We appreciate your comments and suggestions. We have included further discussion in the revised version of this paper, as shown in Lines 451–463.

- Given the limited evidence available for the use of holistic cues or similar, I would remove reference to these / the recommendation to explore these from the main conclusion section within the discussion (and just discuss this only as part of the rest of the main text, as potential avenue for research and application). 

Response: We addressed the issues by relocating the recommendations about holistic focus and analogies from the conclusions section to the discussion, as demonstrated in Lines 417–19, as shown in Lines 417–419.

---

## [Editor Report · Decision Letter 2]

13 Dec 2023

Attentional focus strategies in racket sports: A systematic review

PONE-D-23-11299R2

Dear Dr. Niźnikowski,

We’re pleased to inform you that your manuscript has been judged scientifically suitable for publication and will be formally accepted for publication once it meets all outstanding technical requirements.

Kind regards,

Nick Fogt

Academic Editor

PLOS ONE

Additional Editor Comments (optional):

Thank you for addressing all of the reviewers' concerns.
---

## [Editor Report · Acceptance letter]

19 Dec 2023

PONE-D-23-11299R2 

PLOS ONE

Dear Dr. Niźnikowski, 

I'm pleased to inform you that your manuscript has been deemed suitable for publication in PLOS ONE. Congratulations! Your manuscript is now being handed over to our production team.

Kind regards, 

on behalf of

Dr. Nick Fogt 

Academic Editor

PLOS ONE